# Safety and immunogenicity of a hybrid-type vaccine booster in BBIBP-CorV recipients in a randomized phase 2 trial

Nawal Al Kaabi [1,2,13], Yun Kai Yang[3,13], Li Fang Du[4,5,13], Ke Xu[6,13], Shuai Shao[4,5], Yu Liang[4,5], Yun Kang[5,7], Ji Guo Su[4,5], Jing Zhang[4,5], Tian Yang[3], Salah Hussein[1], Mohamed Saif ElDein[1], Sen Sen Yang[5,7], Wenwen Lei[6], Xue Jun Gao[8], Zhiwei Jiang[9], Xiangfeng Cong[5,7], Yao Tan[5,7], Hui Wang[10], Meng Li[3], Hanadi Mekki Mekki[11], Walid Zaher [12], Sally Mahmoud [12], Xue Zhang[3], Chang Qu[3], Dan Ying Liu[3], Jing Zhang[6], Mengjie Yang[6], Islam Eltantawy[12], Jun Wei Hou[4,5], Ze Hua Lei[4,5], Peng Xiao[12], Zhao Nian Wang[3], Jin Liang Yin[3], Xiao Yan Mao[8], Jin Zhang[10], Liang Qu[3], Yun Tao Zhang[3✉], Xiao Ming Yang [3✉], Guizhen Wu [6✉] & Qi Ming Li [4,5✉]

NVSI-06-08 is a potential broad-spectrum recombinant COVID-19 vaccine that integrates the antigens from multiple SARS-CoV-2 strains into a single immunogen. Here, we evaluate the safety and immunogenicity of NVSI-06-08 as a heterologous booster dose in BBIBP-CorV recipients in a randomized, double-blind, controlled, phase 2 trial conducted in the United Arab Emirates (NCT05069129). Three groups of healthy adults over 18 years of age (600 participants per group) who have administered two doses of BBIBP-CorV 4-6-month, 7-9-month and >9-month earlier, respectively, are randomized 1:1 to receive either a homologous booster of BBIBP-CorV or a heterologous booster of NVSI-06-08. The incidence of adverse reactions is low, and the overall safety profile is quite similar between two booster regimens. Both Neutralizing and IgG antibodies elicited by NVSI-06-08 booster are significantly higher than those by BBIBP-CorV booster against not only SARS-CoV-2 prototype strain but also multiple variants of concerns (VOCs). Especially, the neutralizing antibody GMT against Omicron variant induced by heterologous NVSI-06-08 booster reaches 367.67, which is substantially greater than that boosted by BBIBP-CorV (GMT: 45.03). In summary, NVSI-06-08 is safe and immunogenic as a booster dose following two doses of BBIBP-CorV, which is immunogenically superior to the homologous boost with another dose of BBIBP-CorV.

[1] Sheikh Khalifa Medical City, SEHA, Abu Dhabi, United Arab Emirates. [2] College of Medicine and Health Sciences, Khalifa University, Abu Dhabi, United Arab Emirates. [3] China National Biotec Group Company Limited, Beijing, China. [4] The Sixth Laboratory, National Vaccine and Serum Institute (NVSI), Beijing, China. [5] National Engineering Center for New Vaccine Research, Beijing, China. [6] National Institute for Viral Disease Control and Prevention, Chinese Center for Disease Control and Prevention (China CDC), Beijing, China. [7] Clinical Medicine Office, National Vaccine and Serum Institute (NVSI), Beijing, China. [8] Lanzhou Institute of Biological Products Company Limited, Lanzhou, China. [9] Beijing Key Tech Statistical Consulting Co., Ltd, Beijing, China. [10] Beijing Institute of Biological Products Company Limited, Beijing, China. [11] Union 71, Abu Dhabi, United Arab Emirates. [12] G42 Healthcare, Abu Dhabi, United Arab Emirates. [13]These authors contributed equally: Nawal Al Kaabi, Yun Kai Yang, Li Fang Du, Ke Xu. ✉email: zhangyuntao@sinopharm.com; yangxiaoming@sinopharm.com; wugz@ivdc.chinacdc.cn; liqiming189@163.com

Through the great efforts of researchers worldwide, remarkable achievements have been made in developing effective vaccines against coronavirus disease-19 (COVID-19). As of January 31, 2022, a total of 10 vaccines have been authorized by World Health Organization (WHO) for emergency use[1], and 4,084,470,843 individuals (52.1% of the population) in the world have been fully vaccinated[2]. These COVID-19 vaccines provide efficient protection against severe acute respiratory syndrome coronavirus 2 (SARS-CoV-2). However, the virus is continuously evolving, and a number of variants have emerged, some of which acquired immune escape capability[3]. Due to the pandemic of SARS-COV-2 variants and the waning of immunity over time, the reports of breakthrough infections are growing[3–5]. The Technical Advisory Group on COVID-19 Vaccine Composition (TAG-CO-VAC) of WHO has recommended updating the composition of current COVID vaccines to develop multivalent or broad-protective vaccines against SARS-CoV-2 current and even future variants[6].

Guided by structural and computational analysis of the receptor-binding domain (RBD) of SARS-CoV-2 spike protein, we have designed a mutation-integrated trimeric RBD (mutI-tri-RBD) as the antigen of a recombinant COVID-19 vaccine named NVSI-06-08 (Sinopharm)[7]. In mutI-tri-RBD, three heterologous RBDs, derived respectively from the prototype, Beta and Kappa SARS-CoV-2 strain, were connected end to end and co-assembled into a trimeric structure. By this way, mutI-tri-RBD serves as a hybrid antigen that integrates key mutations from multiple SARS-CoV-2 variants into a single protein. Pre-clinical studies have demonstrated that NVSI-06-08 elicited broader immune response against SARS-CoV-2 variants. The hybrid strategy has also been applied to HIV, coronaviruses and influenza vaccine developments, and it has been proved that hybrid-type vaccine not only can improve immune response but also effectively expand the breadth of immunity[8–10].

Due to the waning of vaccine-induced immunity over time, an effective broad-reactive vaccine is needed as a booster dose to strengthen and broaden immunity against SARS-CoV-2 variants in the individuals who have completed a primary vaccination series. The inactivated vaccine BBIBP-CorV made by Sinopharm is one of the COVID-19 vaccines approved by WHO and has been used in many countries[11]. The interim analysis of a phase 3 trial has demonstrated that the efficacy of two doses of BBIBP was 78.1% against symptomatic COVID-19 cases, and the occurrence of serious adverse events was rare. The geometric mean titer (GMT) of neutralizing antibodies was 156.0 on day 14 after two-dose vaccinations[12]. However, several studies have shown that the neutralizing antibodies elicited by BBIBP-CorV wane over time, suggesting the need for booster vaccinations[13,14]. Given that large-scale populations worldwide have already administered two doses of BBIBP-CorV, in this trial, we evaluate the immunogenicity and safety of NVSI-06-08 as a heterologous booster dose, using homologous boost with BBIBP-CorV as control. As an exploratory study, the cross-reactive immunogenicity of the heterologous BBIBP-CorV/NVSI-06-08 prime-booster vaccination against SARS-CoV-2 variants of concerns (VOCs), including Omicron, was also assessed and compared with that of the homologous booster vaccination with BBIBP-CorV to illustrate the superior immunogenicity of NVSI-06-08 as a booster dose.

## Results

**participants**. From Oct. 23 to Nov. 8, 2021, a total of 1833 healthy adults (≥18 years old) were enrolled, in which 1781 (97.16%) were male. These participants were classified into three groups with different prime-boost intervals, i.e., 4–6 months, 7-9 months and >9 months. For each group, participants were randomly assigned to receive either a heterologous boost of NVSI-06-08 or a homologous boost of BBIBP-CorV. Among the enrolled participants, 1800 individuals (97.17% were male) completed booster vaccination, with 600 participants in each group (Fig. 1). All the 1800 participants who had received the booster dose of vaccination were included in Safety Set (SS) for safety analysis. A total of 1678 participants (97.32% were male) who had no protocol deviations from the follow-up visits were included in Per-protocol set (PPS) for baseline analysis and immunogenicity evaluation (Fig. 1). The majority of the participants were Bangladeshis, Indian or Pakistanis. The demographic characteristics were broadly similar between heterologous and homologous booster groups (Table 1 and Supplementary Table 1). The participants in the two groups had similar age, sex, race, height, and weight distributions.

For the participants, baseline IgG concentrations and neutralizing antibody titers were quantified using a chemiluminescence enzyme immunoassay kit and the live-virus neutralization assay, respectively, before booster vaccination. The baseline antibody levels were statistically similar between the participants in heterologous and homologous booster groups. Before booster vaccination, the baseline IgG GMCs were similar in the participants among groups with different prime-boost intervals. However, neutralizing GMTs in the participants from >9-month group were lower than from 7–9-month group, and those from 7–9-month group were also lower than from 4–6-month group, which demonstrated waning of neutralizing immunity over time (Table 1). The decay of neutralizing immunity highlighted the need for booster vaccination to improve the immune response.

**Safety**. Before initiation of the trial (phase 2), a small-size phase 1 trial was firstly conducted to assess the safety of NVSI-06-08 in healthy adults. In phase 1 study, a total of 48 participants were enrolled, each of whom received three doses of NVSI-06-08 with an interval of one month between each dose. As of Jan 25, 2022, the collected data showed that within 30 days after full vaccinations, 20 (41.67%) participants reported at least one adverse event related to the study vaccine, and all the reported adverse events were grade 1 or 2. No adverse event of grade 3 and above was observed. Adverse events related to the test vaccine were mainly reported in 0-7 days. The most frequent solicited local adverse event related to the test vaccine was pain, and the solicited systemic adverse events were mainly myalgia, headache, and fatigue. No serious adverse event was reported, and no adverse event of special interest occurred. The detailed safety data of phase 1 trial are provided in Supplementary Note 1.

In the phase 2 trial presented here, 146 (16.29%) participants receiving NVSI-06-08 boost and 115 (12.72%) receiving BBIBP-CorV boost reported at least one solicited adverse reaction within 7 days after vaccination, and most of them were of grade 1 or 2. The overall incidence of solicited adverse reactions was low in both booster vaccinations (Fig. 2a, b and Supplementary Table 2). The occurrence of solicited local adverse reactions was quite similar between heterologous and homologous booster groups. All the reported local reactions were of grade 1 or 2, and most of them were injection-site pain (Fig. 2a and Supplementary Table 2). Solicited systemic adverse reactions reported by participants in heterologous booster groups were also similar to those reported by homologous booster groups. The reported systemic reactions were mostly of grade 1 or 2, and the most frequent reactions were headache, muscle pain, fatigue, and fever. Grade 3 systemic reactions, including fever and muscle pain, were only observed in 0.45% participants of heterologous groups and 0.22% participants of homologous group, respectively. No grade 4 or above systemic reaction was found (Fig. 2b and Supplementary

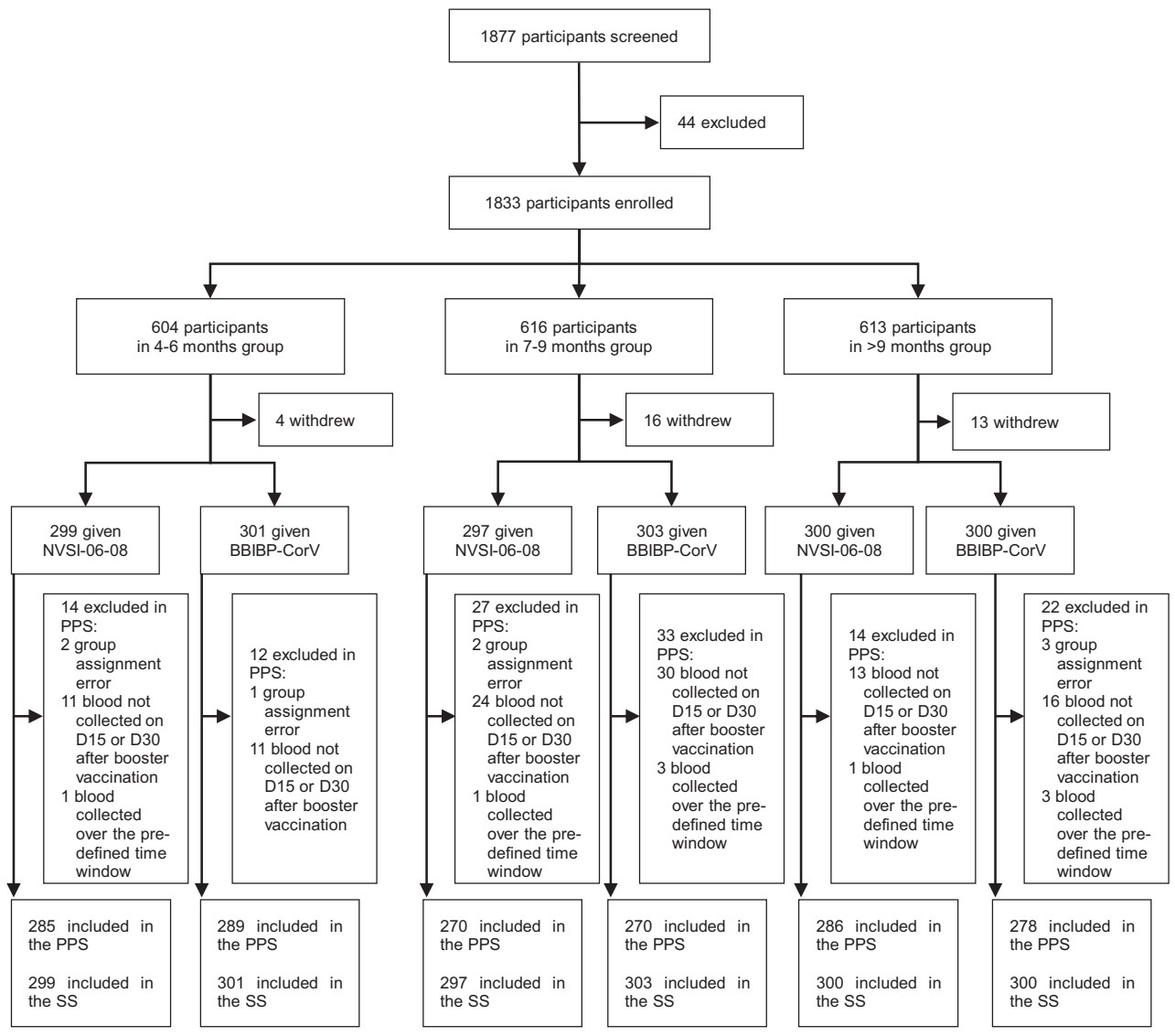

**Fig. 1 Randomization and analysis populations.** A total of 1833 participants were enrolled, and 1800 received booster vaccinations. Participants were classified into three groups with different prime-boost intervals. The participants in each group were randomly assigned to receive a booster dose of eighter NVSI-06-08 or BBIBP-CorV. All the 1800 participants receiving booster vaccination were included in safety set (SS) for safety analysis. A total of 1678 participants who had no protocol deviations on follow-up visits were included in Per-protocol set (PPS) for immunogenicity analysis.

Table 2). The proportion of participants reporting unsolicited adverse reactions was also comparable between heterologous and homologous booster groups (5.36% vs 5.31%). The observed unsolicited reactions primarily included myalgia (0.56% vs 0.66%), fever (0.45% vs 0.55%) and cough (0.33% vs 0.88%), most of which were graded as level 1 or 2 (Supplementary Table 2). In both groups, no AESI and vaccination-related SAE was reported as of the time of this report. Overall, these data suggest that the heterologous boosting with a dose of NVSI-06-08 following two doses of BBIBP-CorV has a good safety profile, which was quite similar to homologous boosting with BBIBP-CorV. The detailed safety data of this trial (phase 2) are provided in Supplementary Note 2.

**Immunogenicity against prototype virus.** According to the study protocol, a total of 23 participants with suspected symptoms were tested by PCR within 30 days after booster vaccination, but all the test results were negative. Immunogenicity analysis showed that both homologous and heterologous booster vaccinations significantly improved the neutralizing antibody titers against the

prototype SARS-CoV-2 virus. However, the post-vaccination neutralizing antibody GMTs of heterologous booster group were dramatically higher than those of homologous booster groups. On day 15 post-vaccination, GMTs of neutralizing antibodies elicited by homologous BBIBP-CorV boost increased by 2.93-fold (95% CI, 2.54–3.37) in 4–6-month group, 10.34-fold (8.78–12.19) in 7–9-month group and 21.44-fold (18.56–24.77) in >9-month group, respectively, compared to the pre-booster baseline levels. Whereas, those elicited by heterologous NVSI-06-08 boost were significantly improved by 40.10-fold (95% CI, 34.61–46.47), 94.42-fold (79.36–112.34), and 246.81-fold (207.02–294.26) in the three groups, respectively (Fig. 3a and Supplementary Table 3). Correspondingly, the fourfold rise rates of neutralizing antibodies boosted by BBIBP-CorV were 22.84% (95% CI, 18.13–28.12%), 75.19% (69.59–80.22%), and 94.24% (90.82–96.67%) in 4–6-month, 7–9-month and >9-month groups, while those boosted by NVSI-06-08 reached 93.68% (95% CI, 90.20–96.21%), 98.15% (95.73–99.40%) and 99.65% (98.07–99.99%) in the three groups, respectively (Fig. 3b and Supplementary Table 3). Neutralizing antibody GMTs and fourfold rise rates further increased on day 30

**Table 1 Demographic characteristics of the participants.**

| | 4–6 months | | | 7–9 months | | | >9 months | | |
|---|---|---|---|---|---|---|---|---|---|
| | NVSI-06-08 (N = 285) | BBIBP-CorV (N = 289) | p value | NVSI-06-08 (N = 270) | BBIBP-CorV (N = 270) | p value | NVSI-06-08 (N = 286) | BBIBP-CorV (N = 278) | p value |
| Age (years) | | | | | | | | | |
| Mean (SD) | 35.47 (8.23) | 34.80 (8.15) | 0.3259 | 34.79 (8.39) | 34.61 (8.25) | 0.8022 | 37.16 (8.29) | 36.86 (7.85) | 0.6572 |
| Median | 35.13 | 33.85 | | 33.84 | 34.13 | | 36.41 | 36.77 | |
| Min, Max | 19.65, 59.84 | 19.60, 64.84 | | 19.16, 66.81 | 18.68, 60.49 | | 18.96, 66.45 | 19.85, 60.85 | |
| Age group, n (%) | | | | | | | | | |
| 18–59 years | 285 (100.00) | 288 (99.65) | 0.3203 | 269 (99.63) | 269 (99.63) | 1.0000 | 284 (99.30) | 277 (99.64) | 0.5794 |
| ≥60 years | 0 (0.00) | 1 (0.35) | | 1 (0.37) | 1 (0.37) | | 2 (0.70) | 1 (0.36) | |
| Sex, n (%) | | | | | | | | | |
| Male | 280 (98.25) | 286 (98.96) | 0.4642 | 261 (96.67) | 257 (95.19) | 0.3839 | 275 (96.15) | 274 (98.56) | 0.0757 |
| Female | 5 (1.75) | 3 (1.04) | | 9 (3.33) | 13 (4.81) | | 11 (3.85) | 4 (1.44) | |
| Height (cm) | | | | | | | | | |
| Mean (SD) | 168.60 (7.59) | 168.69 (6.34) | 0.8861 | 170.51 (7.30) | 169.57 (6.95) | 0.1243 | 169.26 (7.73) | 169.24 (7.70) | 0.9760 |
| Median | 168.00 | 169.00 | | 170.00 | 170.00 | | 170.00 | 170.00 | |
| Min, Max | 124.00, 198.00 | 147.00, 189.00 | | 149.00, 187.50 | 144.40, 187.00 | | 118.00, 189.00 | 106.00, 189.00 | |
| Weight (kg) | | | | | | | | | |
| Mean (SD) | 73.38 (13.18) | 71.86 (11.04) | 0.1346 | 75.25 (13.35) | 74.73 (13.10) | 0.6472 | 75.64 (13.36) | 73.33 (13.00) | 0.0379 |
| Median | 72.00 | 71.30 | | 74.00 | 73.00 | | 74.00 | 72.00 | |
| Min, Max | 43.00, 121.00 | 42.00, 113.00 | | 38.50, 136.00 | 48.00, 117.00 | | 45.00, 156.00 | 45.00, 124.00 | |
| BMI (kg/m$^2$) | | | | | | | | | |
| Mean (SD) | 25.82 (4.46) | 25.28 (3.77) | 0.1136 | 25.85 (4.11) | 25.94 (3.98) | 0.7890 | 26.39 (4.28) | 25.65 (4.75) | 0.0521 |
| Median | 25.51 | 25.10 | | 25.53 | 25.47 | | 25.75 | 25.20 | |
| Min, Max | 16.59, 54.63 | 15.06, 35.16 | | 15.62, 46.51 | 17.73, 39.03 | | 17.63, 50.36 | 15.04, 69.42 | |
| Pre-booster NA GMT (95% CI) | 78.35 (67.10–91.48) | 67.28 (57.41–78.84) | 0.1771 | 51.98 (44.89–60.19) | 48.80 (42.03–56.66) | 0.5526 | 16.96 (14.56–19.74) | 17.47 (14.94–20.44) | 0.7870 |
| Pre-booster IgG GMC (95% CI) | 113.71 (96.45–134.05) | 95.53 (81.53–111.94) | 0.1341 | 133.93 (111.27–161.20) | 118.77 (97.65–144.47) | 0.3810 | 120.45 (96.96–149.63) | 116.05 (91.23–147.62) | 0.8210 |

Results were obtained from the participants who had no protocol deviations.
Comparisons between NVSI-06-08 and BBIBP-CorV booster groups were carried out using Student's *t* test for continuous variables (after log-transformation for antibody titers or concentrations) and Chi-square test for non-ordered categorical variables. All the tests were two-sided and a *p* value < 0.05 was considered statistically significant.
*N* the number of participants, *SD* standard deviation, *BMI* body mass index, *NA* neutralizing antibody, *GMT* geometric mean titer, *GMC* geometric mean concentration.

after booster vaccination, and the neutralizing responses induced by heterologous boost were also remarkably superior to those by homologous boost ($p < 0.0001$). On day 30 post-vaccination, homologous boost of BBIBP-CorV led to 4.38-fold (95% CI, 3.81–5.05), 22.40-fold (19.19–26.15), and 46.26-fold (39.76–53.83) increases from baselines in neutralizing antibody GMTs for participants from 4–6-month, 7–9-month and >9-month groups, respectively, whereas much greater increases of 47.61-fold (95% CI, 41.17–55.06), 148.50-fold (126.60–174.19) and 441.11-fold (373.91–520.38) were obtained by heterologous boost of NVSI-06-08 (Fig. 3a and Supplementary Table 3). A similar increase in trend was also observed in fourfold rise rates. For 4–6-month, 7–9-month and >9-month groups, the fourfold rise rates induced by homologous boost were 38.75% (95% CI, 33.11–44.64%), 92.96% (89.23–95.71%) and 98.20% (95.85–99.41%), respectively, which increased to 96.84% (95% CI, 94.09–98.55%), 99.26% (97.35–99.91%) and 100.00% (98.72–100.00%) induced by heterologous boost (Fig. 3b and Supplementary Table 3). Among three groups with different prime-boost intervals, the post-vaccination neutralizing antibody levels in the participants of 7–9-month group were comparable to those of >9-month group, both of which were significantly higher than those of 4–6-month group. Especially, the neutralizing GMTs elicited by NVSI-06-08 in 7–9-month group reached as high as 7719.35 against wild-type SARS-CoV-2 virus. The results suggest that a booster dose with a prime-boost interval over 6 months is immunogenically optimal.

The immunogenic superiority of heterologous NVSI-06-08 booster to homologous BBIBP-CorV booster was also confirmed by anti-RBD IgG response. In line with the neutralizing antibody titers, both homologous and heterologous booster vaccinations significantly improved the RBD-specific IgG antibody levels. However, heterologous booster induced dramatically higher IgG GMCs than homologous booster in all groups with different prime-boost intervals. Compared to pre-booster baselines, anti-RBD IgG GMCs increased by 2.56–3.58-fold at 15 days after vaccination in the groups boosted with BBIBP-CorV, whereas, much higher 49.15–62.62-fold increases were observed in the groups boosted with heterologous NVSI-06-08 (Fig. 4a and Supplementary Table 4). Similarly, fourfold rise rates elicited by heterologous booster were significantly higher than those by homologous booster in all groups with different prime-boost intervals (90.56–96.14% vs 20.76–31.65%, $p < 0.0001$) (Fig. 4b and Supplementary Table 4). At 30 days after booster vaccination, similar results were obtained. IgG antibody GMCs boosted by NVSI-06-08 increased from baseline by 44.24-fold (95% CI, 37.82–51.75) in the 4–6-month group, 36.94-fold (30.65–44.52) in the 7–9-month group and 41.18-fold (33.25–51.01) in >9-month group, respectively, which were remarkably greater than 2.24-fold (95% CI, 1.94–2.58), 2.45-fold (2.04–2.93) and 2.31-fold (1.89–2.82) boosted by BBIBP-CorV ($p < 0.0001$) (Fig. 4a and Supplementary Table 4). Correspondingly, the fourfold rise rates induced by heterologous booster were 95.44% (95% CI, 92.33–97.55%), 92.22% (88.36–95.12%), and 88.11% (83.79–91.62%), which were much higher than 17.30% (13.12–22.16%), 22.96% (18.08–28.45%) and 21.94% (17.22–27.27%) induced by homologous booster (Fig. 4b and Supplementary Table 4).

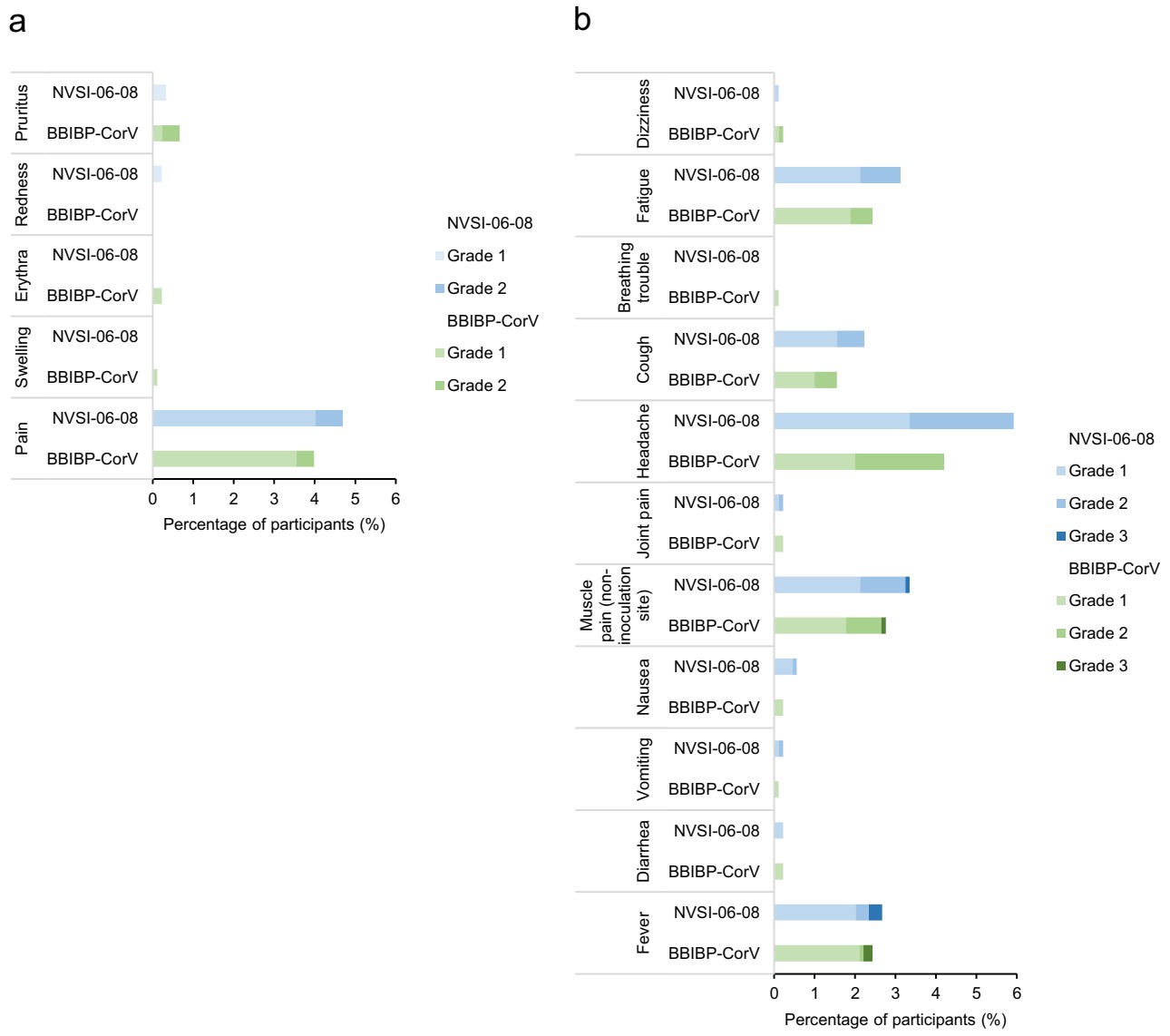

**Fig. 2 Incidence and severity of solicited adverse reactions after booster vaccinations with NVSI-06-08 and BBIBP-CorV, respectively. a, b** Incidence and severity of local (**a**) and systemic (**b**) adverse reactions after boosted with NVSI-06-08 were compared to those boosted with BBIBP-CorV. Adverse reactions are graded according to the relevant guidance of the China National Medical Products Administration (NMPA).

**Neutralizing antibody response against Omicron and other VOCs.** NVSI-06-08 was designed as a hybrid-type vaccine with broader neutralizing profiles, the cross-reactive immunogenicity of heterologous NVSI-06-08 booster against multiple SARS-CoV-2 VOCs, including Omicron, was evaluated as an exploratory study. A total of 200 serum samples, collected on day 15 after booster vaccination, from the participants with sequential enrollment numbers in the 7–9-month group (99 participants receiving heterologous boost and 101 receiving homologous boost) were used in the cross-neutralizing activity tests.

Both in homologous and heterologous booster groups, the neutralizing antibody level against SARS-CoV-2 Omicron variant was significantly reduced in comparison with that against the prototype strain, indicating the distinct immune-evasive ability of Omicron variant. Our results are consistent with other studies[15–18]. However, the anti-Omicron neutralizing titers induced by heterologous boost were still significantly higher than those induced by homologous boost. In participants receiving the homologous boost of BBIBP-CorV, the neutralizing antibody

GMT against Omicron variant was 45.03 (95% CI, 36.37–55.74). By comparison, in participants boosted by the heterologous NVSI-06-08, the anti-Omicron neutralizing GMT still maintained at a high level of 367.67 (95% CI, 295.50–457.47), which was 8.17-fold higher than that induced by homologous boost (Fig. 5 and Supplementary Table 5).

We also evaluated the neutralizing antibody response against other several SARS-CoV-2 VOCs, including Alpha, Beta, and Delta. All the tested VOCs were significantly less sensitive to the neutralization induced by BBIBP-CorV booster. However, the neutralizing antibody response against Alpha and Beta variants offered by NVSI-06-08 booster was comparable to that against prototype strain. For all the tested variants, heterologous booster induced substantially greater neutralizing antibody levels than homologous booster. The GMTs of neutralizing antibodies boosted by NVSI-06-08 were 11.04-fold, 14.98-fold, and 9.48-fold higher than those boosted by BBIBP-CorV against Alpha, Beta, and Delta variants, respectively (Fig. 5 and Supplementary Table 5).

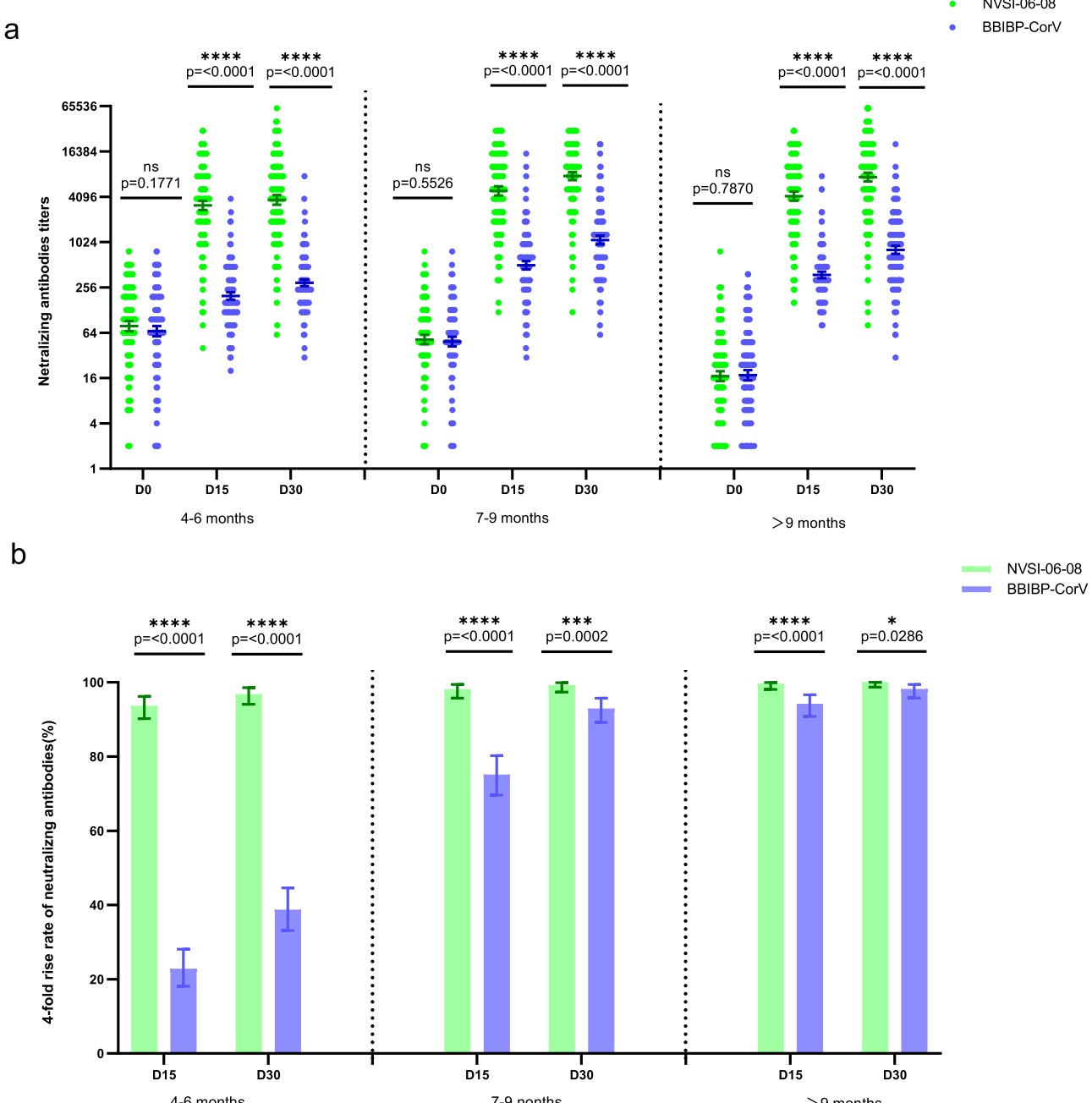

**Fig. 3 Neutralizing antibody levels against prototype SARS-CoV-2 before and 15 and 30 days after the booster vaccinations. a** GMTs of neutralizing antibodies increased from baseline (day 0) to day 15 and 30 post-boosting elicited by heterologous NVSI-06-08 booster, compared with those induced by homologous BBIBP-CorV booster. **b** Correspondingly, the four-fold rise rates of neutralizing antibodies on day 15 and 30 after boosting elicited by NVSI-06-08 booster, compared with those induced by BBIBP-CorV booster. Both in **a** and **b**, for NVSI-06-08 booster vaccination, $n = 285$ in 4–6-month group, $n = 270$ in 7–9-month group, and $n = 286$ in >9-month group. For BBIBP-CorV booster vaccination, $n = 289$ in 4–6-month group, $n = 270$ in 7–9-month group, and $n = 278$ in >9-month group. Data are presented as GMTs and 95% CIs. Differences in neutralizing antibody titers between heterologous and homologous booster groups were tested with two-sided grouped t-test after log transformation. The four-fold rise rates between heterologous and homologous booster groups were compared by two-sided Fisher's exact test. A two-sided $p$ value < 0.05 was considered significant. *$p < 0.05$, ****$p < 0.0001$. The exact $p$ values are presented in the figure.

## Discussion

Findings from this trial show that the heterologous prime-boost vaccination with one dose of NVSI-06-08 following two doses of BBIBP-CorV was safe, tolerant, and immunogenic in healthy adults. The heterologous prime-boost regimen with BBIBP-CorV/NVSI-06-08 was immunogenically superior to homologous BBIBP-CorV boost. The neutralizing antibody GMTs elicited by heterologous boost were 9.72–15.96-fold higher than those elicited by homologous boost (GMT, 3141.92–4908.34 versus 196.89–504.76) on day 15 post-boosting, and 7.06–12.65-fold higher (GMT, 3730.18–7719.35 versus 294.96–1093.26) on day 30 post-boosting. Compared to the peak value of neutralizing antibody level (GMT, 282.7) after priming vaccination with two doses of BBIBP-CorV as reported in the previous study[19],

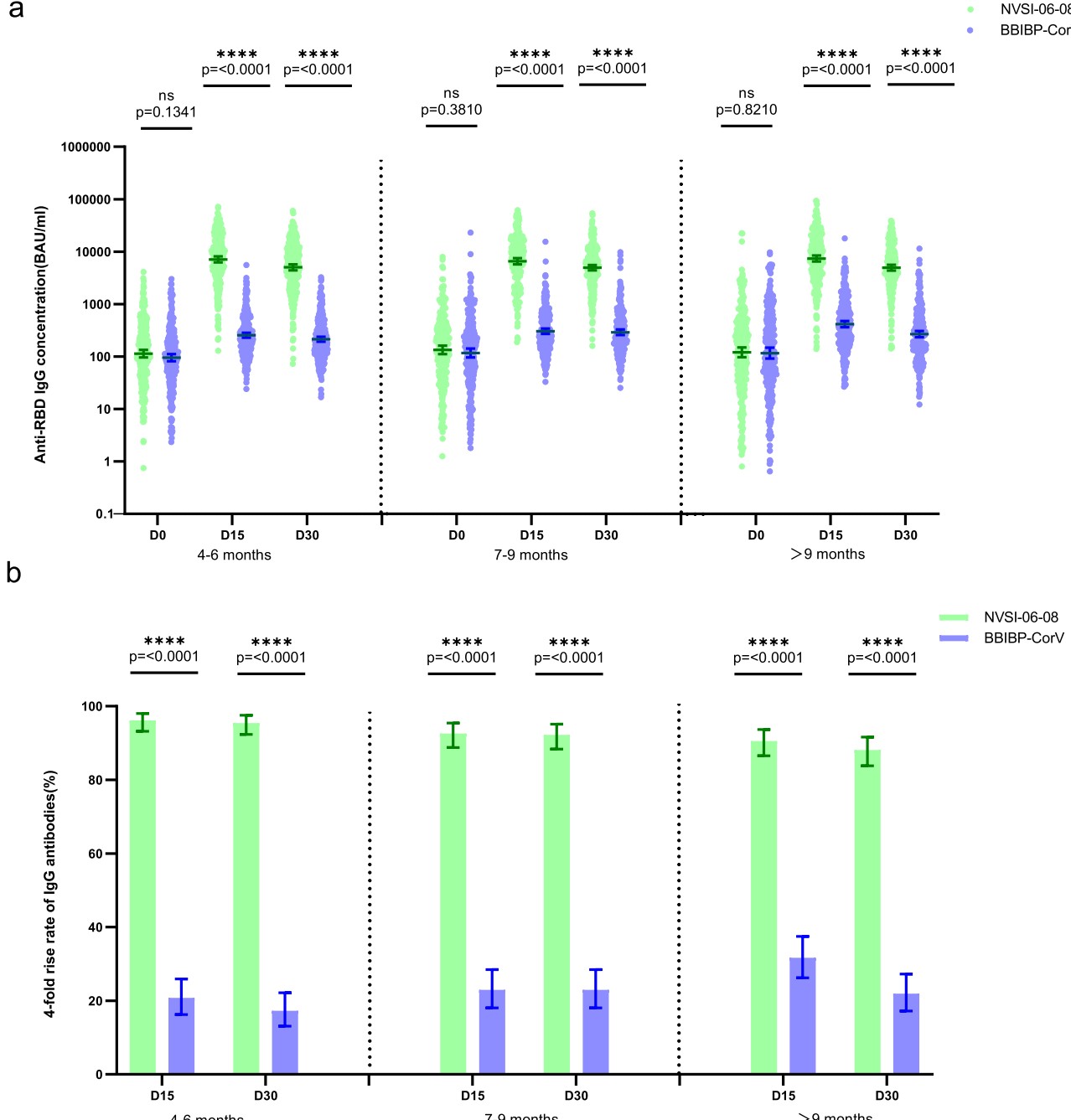

**Fig. 4 RBD-binding IgG antibody levels against prototype SARS-CoV-2 before and 15 and 30 days after the booster vaccinations. a** GMCs of RBD-binding IgG antibodies increased from baseline (day 0) to day 15 and 30 post-boosting elicited by heterologous NVSI-06-08 booster, compared with those induced by homologous BBIBP-CorV booster. **b** Correspondingly, the fourfold rise rates of IgG antibodies on days 15 and 30 after boosting elicited by NVSI-06-08 booster, compared with those induced by BBIBP-CorV booster. Both in **a** and **b**, for NVSI-06-08 booster vaccination, $n = 285$ in 4–6-month group, $n = 270$ in 7–9-month group, and $n = 286$ in >9-month group. For BBIBP-CorV booster vaccination, $n = 289$ in 4–6-month group, $n = 270$ in 7–9-month group, and $n = 278$ in >9-month group. Data are presented as GMCs and 95% CIs. Differences in RBD-binding IgG antibody concentrations between heterologous and homologous booster groups were tested with two-sided grouped $t$ test after log transformation. The fourfold rise rates between heterologous and homologous booster groups were compared by two-sided Fisher's exact test. A two-sided $p$ value < 0.05 was considered significant. *$p$ < 0.05, ****$p$ < 0.0001. The exact $p$ values are presented in the figure.

the neutralizing GMT was improved 1.04–3.87-fold on day 30 after a booster dose of BBIBP-CorV, and more remarkably 13.19–27.31-fold after a heterologous booster dose of NVSI-06-08. Multiple lines of evidence have demonstrated that neutralizing antibody titers are highly correlated with protective efficacy against the infection of SARS-CoV-2[20–22]. The improvement of neutralizing antibody GMT is an indicator of enhancement of

protective efficacy offered by the vaccine. Our findings indicate that a heterologous boost with NVSI-06-08 following prime vaccination with BBIBP-CorV could provide stronger protection against SARS-CoV-2 than the third dose of BBIBP-CorV.

The incidence of adverse reactions was low in both heterologous and homologous booster vaccinations, and most of the reported local and systemic adverse reactions were of grade 1 or 2.

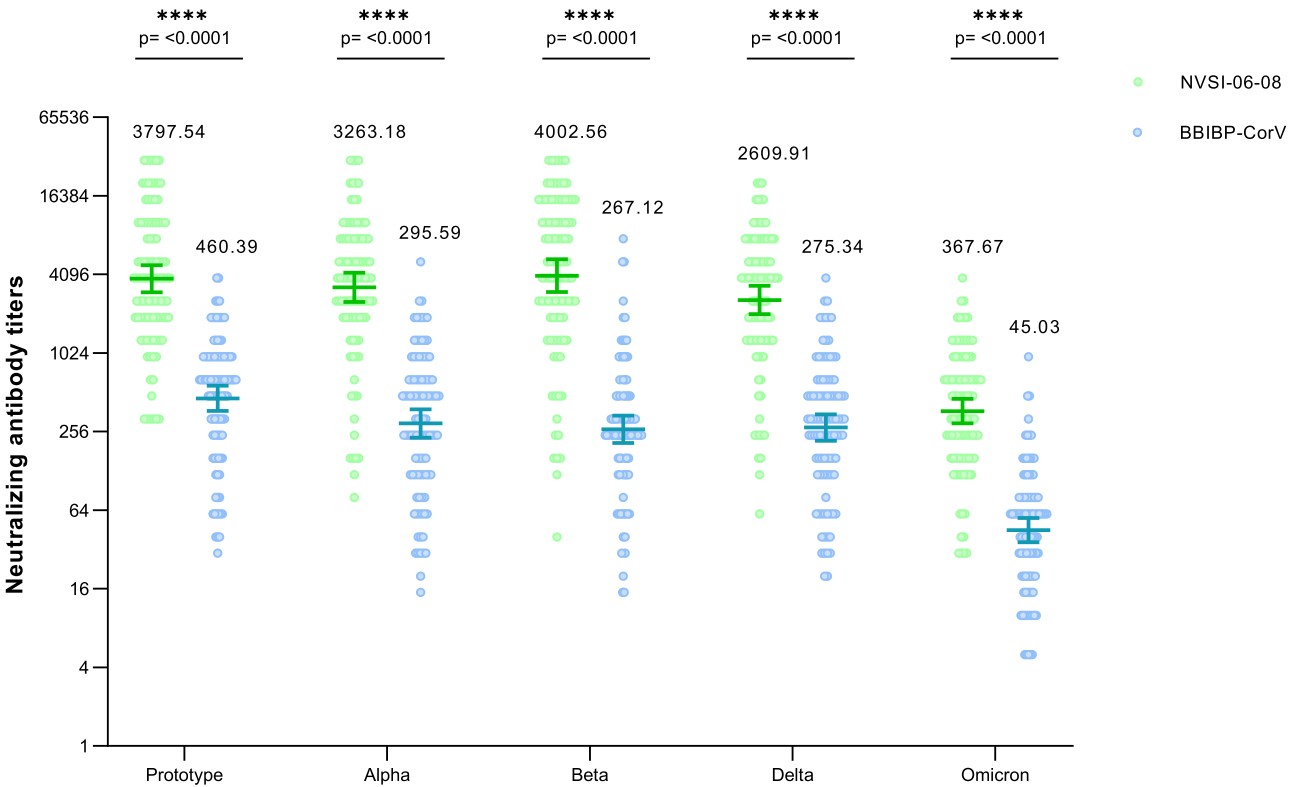

**Fig. 5 Cross-neutralizing antibody titers against SARS-CoV-2 prototype stain and several VOCs, including Alpha, Beta, Delta, and Omicron, elicited by heterologous NVSI-06-08 booster, compared with those elicited by homologous BBIBP-CorV booster.** A subset of 200 serum samples, collected on day 15 post-boosting, from the participants with sequential enrollment numbers in 7–9-month group (99 participants receiving a booster dose of NVSI-06-08 and 101 receiving a third dose of BBIBP-CorV) were tested using live-virus neutralization assay. Data are presented as GMTs and 95% CIs. The GMT values are given on the graph. Differences between NVSI-06-08 and BBIBP-CorV booster groups were tested with two-sided grouped $t$ test after log transformation. A two-sided $p$ value < 0.05 was considered significant. ****$p$ < 0.0001. The exact $p$ values are presented in the figure.

The overall safety profile of heterologous boost was quite similar to that of homologous boost, which was also comparable to the safety of the priming with two doses of BBIBP-CorV as reported previously[19]. A booster dose did not distinctly increase the risk of more serious side effects. The antigen of NVSI-06-08 was designed without introducing any exogenous sequence, which did not bring additional safety risks. In the vaccine, aluminum adjuvant was used, whose safety has been verified for a long time. All these features contributed to the high safety profile of NVSI-06-08.

NVSI-06-08 was designed as a hybrid-type COVID-19 vaccine with broad protection potential, which integrated multiple antigens into a single molecule. Our studies show that booster vaccination of NVSI-06-08 elicited a potent cross-neutralizing response against various SARS-CoV-2 VOCs. Especially, the neutralizing activity induced by NVSI-06-08 booster against the highly immune-evasive Beta variant was no less than that against the prototype strain, which demonstrated the immunological superiority of hybrid-type vaccine. Although the immunity offered by NVSI-06-08 booster was less effective against Omicron variant as compared to that against the prototype virus, the anti-Omicron neutralizing GMT still maintained at a considerable level of 367.67 (95% CI, 295.50–457.47) on day 15 after booster vaccination, which was substantially higher than that boosted by BBIBP-CorV. According to the increasing trends in immune response, a greater anti-Omicron neutralizing GMT can be obtained on day 30 post-boost. Given that an Omicron-specific vaccine is not yet available, the prime-boost vaccination with inactivated COVID-19 vaccine plus NVSI-06-08 may be an optional strategy against the pandemic of Omicron. The RBD of

Omicron harbors numerous mutations, most of which were not integrated into the antigen of NVSI-06-08. We believe that updating the vaccine by including Omicron-carrying mutations into the immunogen should induce better immunogenicity against Omicron variant. Previous studies have indicated that hybrid-type immunogen could elicit superior B cell responses both in quantity and quality in comparison with the homologous immunogens. The superior immunogenicity of heterologous immunogen may be attributed to the avidity advantage to cross-reactive B cells[10].

Considering that inactivated COVID-19 vaccines have been inoculated in large-scale populations worldwide, and the immunity offered by the vaccine decays over time[13,14], it is of significance to choose a preferred vaccine as a booster dose to restore and even enhance the immunity against SARS-CoV-2. Our studies provided a quite effective booster strategy for the inactivated vaccine recipients. The high level of neutralizing antibodies induced by heterologous BBIBP-CorV/NVSI-06-08 vaccination could alleviate the waning of immunity and facilitate the formation of an immune barrier, which then may suppress virus mutations.

The immunogenicity data show that both the absolute value and the fold rise of post-booster neutralizing antibody titers in the 7-9-month and >9-month groups were significantly greater than those in the 4-6-month group. Especially, although the pre-booster baseline in the >9-month group has waned more, the post-booster neutralizing titers were still distinctly higher than those in the 4-6-month group. Our results indicate that the prime-boost vaccination with an interval over 6 months was immunogenically superior to the interval of 4-6 months.

Many studies have revealed that prime-boost interval has a significant influence on the immune efficacy of COVID vaccines[23,24]. Our results are consistent with those previously reported studies. An over-6-month interval may facilitate better maturation of memory cells, and improve binding affinity and production level of antibodies.

This trial has several limitations. Firstly, among participants, the proportion of elderly persons aged ≥ 60 years was low, and the immune response of the BBIBP-CorV/NVSI-06-08 booster strategy in the older population should be further assessed in the future. Secondary, the number of male participants were much larger than female, and the data may not well reflect the effect in women. Thirdly, in this trial, PCR tests were conducted only for the subjects who showed any symptoms of COVID-19 or went to the hospital for treatment. It may be better to perform PCR tests at regular intervals for all the participants to monitor the incidence of COVID-19 infections after booster vaccination, which will be carried out in the following phase 3 trial.

In summary, heterologous prime-boost vaccination with BBIBP-CorV plus NVSI-06-08 was well tolerated and immunogenic against not only SARS-CoV-2 prototype strain but also the VOCs including Omicron. It was immunogenically superior to the booster with the third dose of BBIBP-CorV. The findings also implied that hybrid-type vaccine could induce potent and broad immune activities, which may provide an effective strategy for broad-spectrum vaccine developments.

## Methods

**Trial design and participants**. We conducted a randomized, double-blind, controlled phase 2 trial in the United Arab Emirates (NCT05069129) to evaluate the immunogenicity and safety of NVSI-06-08 (Sinopharm) as a booster dose following a primary series of BBIBP-CorV (Sinopharm). From Oct 23 to Nov 8, 2021, participants were recruited. Trial participants included three groups of healthy adults aged ≥18 years who had received two doses of BBIBP-CorV 4–6 months, 7–9 months, and >9 months before, respectively. Written informed consent was obtained from all participants before the screening. Participants were enrolled after undergoing a health screening by inquiry, medical history review, and physical examination. Confirmed, suspected or asymptomatic COVID-19 cases were excluded from the trial. Individuals with a history of SARS or MERS infections and those vaccinated with any other COVID-19 vaccines were also excluded. The detailed inclusion and exclusion criteria are listed in the study protocol (Supplementary Note 3).

**Trial oversight**. The trial protocol was reviewed and approved by Abu Dhabi Health Research and Technology Ethics Committee. The trial was conducted in accordance with Good Clinical Practice (GCP), the Declaration of Helsinki (with amendments) as well as the local legal and regulatory requirements, and trial safety was overseen by an independent safety monitoring committee. China National Biotec Group Co., Ltd. (CNBG) of Sinopharm was the regulatory sponsor of the trial. The trial was funded by Lanzhou Institute of Biological Products Co., Ltd (LIBP) of Sinopharm and Beijing Institute of Biological Products Co., Ltd (BIBP) of Sinopharm. National Vaccine and Serum Institute (NVSI) of Sinopharm and China National Biotec Group Co., Ltd. (CNBG) of Sinopharm designed the trial, performed the analyses, and interpreted the data. All authors had full access to study data and the corresponding authors were responsible for the decision to submit the manuscript.

**Studied vaccines**. NVSI-06-08 is a potential broad-spectrum recombinant COVID-19 vaccine, using a hybrid mutI-tri-RBD as the antigen[7], which was developed by the National Vaccine and Serum Institute (NVSI) of Sinopharm and manufactured by the Beijing Institute of Biological Products Co., Ltd (BIBP) of Sinopharm and Lanzhou Institute of Biological Products Co., Ltd (LIBP) of Sinopharm in accordance with good manufacturing practice (GMP). One dose of NVSI-06-08 contains 20 μg antigen and 0.3 mg aluminum hydroxide adjuvant. BBIBP-CorV is an inactivated vaccine produced by the Beijing Institute of Biological Products Co., Ltd (BIBP) of Sinopharm, which has been approved by WHO for emergency use and applied in many countries worldwide[11].

**Randomization and blinding**. For allocation of the participants, a randomization list was created by the stratified blocked randomization method using the SAS software (version 9.4). Eligible participants were stratified according to the prime-boost intervals, i.e., 4–6 months, 7–9 months, and >9 months. In each stratum, participants were randomly assigned in a ratio of 1:1 to either heterologous or homologous booster groups using a block randomization method with a block size of 10. A vaccine randomization list was also generated using SAS software with a randomization block size of 10. Then, participant and vaccine randomization lists were inputted into the interactive web response system (IWRS), and participants were vaccinated according to the randomization number and vaccine number obtained from IWRS. The randomization and blinding were conducted by independent personnel who were not involved in the study. Participants, investigators, and other staff remained blinded to individual treatment assignments during the trial.

**Procedures**. At the trial site, each eligible participant received a booster vaccination of NVSI-06-08 or BBIBP-CorV through intramuscular injection in the deltoid muscle of the upper arm. After booster vaccination, participants were observed at the study site for 30 mins to identify immediate adverse reactions. Solicited adverse events (AEs) were recorded for 7 days and unsolicited AEs for 30 days post-vaccination. Serious adverse events (SAEs) and adverse events of special interest (AESIs) were collected up to 6 months after a full course of immunization. The grades of local and systemic adverse events were determined according to the relevant guidance of the China National Medical Products Administration (NMPA). Nasopharyngeal swabs were collected from all the participants for PCR tests prior to booster vaccination. After the vaccination, only for the subjects who showed any symptoms of COVID-19 or went to the hospital for treatment, nasopharyngeal swabs were collected and PCR tests were conducted. Blood samples were collected before booster vaccination, and on 15 days, 30 days, 3 months, 6 months, 9 months, and 12 months after booster vaccination. Vaccination and blood collection were conducted at SKMC Center for Diabetes & Endocrinology, Abu Dhabi, UAE. The data were collected in accordance with local regulations and ICH-GCP relevant standards.

**Study outcomes**. The primary immunogenicity outcome was the neutralizing response on 15 days and 30 days after booster vaccination, by evaluation of the geometric mean titers (GMTs) of neutralizing antibodies and the corresponding fourfold rise rate (i.e., post-/pre-boost ≥4) against SARS-CoV-2 prototype strain. Neutralizing antibody titers were measured using live-virus neutralization assay. The secondary immunogenicity outcome was geometric mean concentrations (GMCs) of IgG antibodies and the corresponding four-fold rise rate against SARS-CoV-2 prototype strain. IgG antibodies were measured using a magnetic particle-based chemiluminescence enzyme immunoassay kit. The safety outcome was the occurrence and severity of any adverse reactions within 30 days post-boost. As an exploratory study, the immunogenicity of booster vaccination against SARS-CoV-2 variants of concerns (VOCs), including Omicron, was also evaluated by the GMTs of neutralizing antibodies in a subset of participants from 7–9-month group. Neutralizing antibody titers against the VOCs, including Alpha, Beta, Delta, and Omicron, were measured using live-virus neutralization assay.

**Laboratory analyses**. Spike receptor-binding domain (RBD)-specific IgG antibodies against the prototype SARS-CoV-2 strain were measured using a commercially available magnetic particle-based chemiluminescence enzyme immunoassay kit purchased from Bioscience (Chongqing) Biotechnology Co. (approved by the China National Medical Products Administration; approval numbers 20203400183). The IgG antibody detections were carried out on an automated chemiluminescence detector (Axceed 260) according to the manufacturer's instructions. The reference calibrator used in the kit can be traced back to WHO International Standard First WHO International Standard for anti-SARS-CoV-2 immunoglobulin (human) NIBSC code: 20/136.

Neutralizing antibody titers against prototype strain and VOCs, including Alpha, Beta, Delta, and Omicron, were evaluated using live-virus neutralization assay. Serum samples were heat-inactivated at 56 °C for 30 mins, and then serially diluted by twofold starting from 1:4 (in the detection of neutralizing antibodies against prototype SARS-CoV-2 strain) or 1:10 (in the detection of neutralizing antibodies against VOCs) dilution. The diluted serum was mixed with an equal volume of 100 TCID$_{50}$ of SARS-CoV-2 live virus and incubated at 37 °C for 2 h. After that, the Vero cell suspension with a density of $1.5–2 \times 10^5$ cells per mL was added to the serum-virus mixture, and then the plates were incubated at 37 °C for 3–5 days. Both cell-only and virus-only wells were also set as controls. Neutralizing antibody titer was determined as the reciprocal of the serum dilution for 50% protection against viral infection to the cell. The titer for the serum below the limit of detection was set to half value of the detection limit. The live-virus neutralization assays were performed in the BSL3 facility of the National Institute for Viral Disease Control and Prevention, Chinese Center for Disease Control and Prevention (China CDC), Beijing, China. SARS-CoV-2 live viruses of the prototype (QD-01), Alpha (BJ-210122-14), Beta (GD84), Delta (GD96), and Omicron (NPRC2.192100003) strains were used in the assays. All these viruses were obtained from the National Institute for Viral Disease Control and Prevention, the Chinese Center for Disease Control and Prevention (China CDC). The Vero cell used in the assay was obtained from the National Institute for Food and Drug Control (NIFDC) of China. Both RBD-binding IgG detection and live-virus neutralization assays were carried out in a blinded manner.

**Statistical analysis**. The sample size of participants was determined by Power Analysis and Sample Size (PASS15.0) software using the expected difference between groups, predefined non-inferiority margin, intended power, significance level, and estimated dropout rate. Assuming that the fourfold rise rate after booster vaccination achieves 85%, 208 participants in each arm will be required to have 80% power to conclude non-inferiority with margin of −10% and one-sided significance level of 2.5% using Miettinen and Nurminen method. If equal GMT after booster immunization is assumed, and the standard deviation of GMT after log10 transformation is considered to be 0.7, 250 subjects in each arm will be required to have 80% power to conclude non-inferiority with a margin of 2/3 and a one-sided significance level of 2.5%. Then, considering about 15–20% drop-out rate, 600 participants are required in each of the three groups (4–6-month, 7–9-month, and >9-month groups), and 1800 subjects in total are planned to enroll.

All statistical analyses were carried out using the SAS 9.4 software (SAS Institute, Cary, NC). Baseline characteristics were analyzed based on the participants who had no protocol deviations. Two-sided Student's $t$ test and two-sided Chi-square test were used, respectively, for the comparison of continuous and categorical characteristics between groups. Safety was analyzed based on the safety set (SS) that included the participants receiving booster vaccination. Safety analysis results were presented as counts and percentages of adverse reactions. The differences in safety between heterologous and homologous booster groups were analyzed using two-sided Fisher's exact test. Immunogenicity was evaluated based on the Per-protocol set (PPS), which included the participants without protocol deviation, and both the GMCs of RBD-specific IgG and GMTs of neutralizing antibodies were computed along with 95% CIs. According to the pre-booster and post-booster values, the fold rises in IgG GMCs and neutralizing antibody GMTs, as well as the associated 95% CIs, were calculated. In addition, the fourfold rise rates of IgG and neutralizing antibodies along with the associated Clopper–Pearson 95% CIs were also computed. RBD-specific IgG concentrations and neutralizing antibody titers between heterologous and homologous booster groups were compared using two-sided grouped $t$ test after log transformation. The fold rises in IgG GMC and neutralizing GMT between two groups were also compared using two-sided grouped $t$ test. Fourfold rise rates between heterologous and homologous booster groups were compared by two-sided Fisher's exact test.

**Reporting summary**. Further information on research design is available in the Nature Research Reporting Summary linked to this article.

## Data availability

The study protocol is available in the Supplementary Information file. The individual participant data will be shared after de-identification. The clinical trial is still ongoing, and the data will be available from 1 month to 1 year after the completion of the study. Researchers who provide a scientifically sound proposal will be allowed access to the individual participant data. Proposals should be directed to the corresponding author Qi Ming Li (liqiming189@163.com). The proposals will be reviewed and approved by the funder, investigator, and collaborators on the basis of scientific merit. To gain access, data requestors will need to sign a data access agreement.

## Code availability

All codes that produced the results are available upon request to the corresponding author Qi Ming Li (liqiming189@163.com) with a scientifically sound proposal.

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

## Acknowledgements

This study was sponsored by China National Biotec Group Co., Ltd. (CNBG) of Sinopharm and funded by Lanzhou Institute of Biological Products Co., Ltd (LIBP) of Sinopharm and Beijing Institute of Biological Products Co., Ltd (BIBP) of Sinopharm. The sponsor contributed to trial design, data analyses, and data interpretation. The funder had no role in trial design, data collection, data analysis, data interpretation, or writing of the paper. We thank Prof. Guoyong Yuan from the University of Hong Kong for providing SARS-CoV-2 Omicron virus used in cross-reactive neutralizing antibody detection assays.

## Author contributions

N.A.K. was the chief investigator. N.A.K., Q.M.L., Y.T.Z., X.M.Y., Y.K.Y., Jing Zhang (NVSI), and Y.K. designed the trial and study protocol. Q.M.L., Jing Zhang (NVSI), Y.L., and J.G.S. designed the recombinant vaccine NVSI-06-08. Q.M.L., Y.T.Z., and X.M.Y. were responsible for the organization and supervision of the project. Q.M.L., Jing Zhang (NVSI), Y.L., X.J.G., and X.Y.M. provided the NVSI-06-08 vaccine. H.W. and Jin Zhang. provided the inactivated vaccine BBIBP-CorV. X.M.Y., Y.T.Z., I.E., and Y.K.Y. contributed to project management. T.Y., Y.K., M.L., L.Q., W.Z., P.X., X.Z., C.Q., D.Y.L., and S.S.Y. participated in the implementation of the trial. H.M.M., Z.N.W., and J.L.Y. conducted sample collection and processing. G.W., K.X., W.L., Jing Zhang (China CDC), M.Y., and S.M. carried out the serum tests. G.W., K.X., Y.L. L.F.D., J.W.H., and Z.H.L. contributed to the development of the serum testing method and manuscript preparation. S.H. and M.S.E. performed the clinical data gathering, analysis, and operation. Z.J., X.C., and Y.T. performed statistical analysis of the data. Q.M.L., Jing Zhang (NVSI), Y.L., G.W., Y.K.Y., and K.X. performed data analysis and interpretation. J.G.S., Jing Zhang (NVSI), L.F.D., Y.L., K.X., S.S., and S.S.Y. contributed to the writing of the manuscript. Q.M.L. and G.W. oversaw the final manuscript preparation. All authors approved the final manuscript.

## Competing interests

Y.K.Y., T.Y., M.L., X.Z., C.Q., D.Y.L., Z.N.W., J.L.Y., L.Q., Y.T.Z., and X.M.Y. are employees of the China National Biotec Group Company Limited. L.F.D., S.S., Y.L., Y.K., J.G.S., Jing Zhang (NVSI), S.S.Y., X.C., Y.T., J.W.H., Z.H.L., and Q.M.L. are employees of the National Vaccine and Serum Institute (NVSI). X.J.G. and X.Y.M. are employees of Lanzhou Institute of Biological Products Company Limited (LIBP). H.W. and Jin Zhang are employees of Beijing Institute of Biological Products Company Limited (BIBP). L.F.D., S.S., Y.L., J.G.S., Jing Zhang (NVSI), J.W.H., Z.H.L., and Q.M.L. are listed as inventors of the patent applications for the recombinant trimeric RBD-based vaccines (Application numbers: 202110348881.6, 202110464788.1 and 202110676901.2). The other authors declare no competing interests.
