## [Peer review file · Nature Communications]

REVIEWERS' COMMENTS

Reviewer #1 (Remarks to the Author):

Page 10 line 206, these seem like comparison of prototype neutralization versus vs omicron neutralization within each arm (homologous/heterologous). Seems more natural to emphasize the omicron versus prototype difference between arms.

Page 13 line 282. In the discussion you talk about how the benefit of boosting is greater >6 month group compared to the 4-6 month group, presumably from Figure 3a. But I think this is a comparison of increases relative to baseline, and the >9 month group has waned more. The actual D30 neuts look similar in the three groups. Some discussion of this point seems warranted.

Reviewer #2 (Remarks to the Author):

The authors offer a well described study comparing NVS1-06-08 boosting with homologous BBIBP-CoV boosting and give a clear outline of the recruitment, demographics and outcomes. It is a nicely written manuscript and provides valuable insight into new vaccines that induce cross-reactive immunity.

I only have minor transcriptional comments

Line 140 Change "Significantly" to "significant increases"

Line 142 change "boosting" to "booster" - change boosting to booster throughout

Line 145 Change "were increased" to "increased by"

Line 148 Change "improved" to "significantly".

Fig 3 Neutralising data shows increasing fold rise following the homologous schedule, particularly with the longer intervals. However, with the total IgG (fig 4) there is no change in fold rise with the longer intervals. I wonder if the authors plan to look at the immunoglobulin Fc to see whether this changes over time to account for increasing function (neutralisation) even though total IgG doesn't increase as dramatically. It would be interesting further investigation.

Reviewer #3 (Remarks to the Author):

The authors submit a manuscript on a randomized, double blind, controlled trial evaluating a recombinant vaccine incorporating three heterologous RBDs (NVTI-06-080) as a booster vaccine in comparison with homologous booster in individuals who received an initial vaccination series with BBIBP-CorV, an inactivated COVID-19 vaccine. Participants in this study were randomized to either heterologous boost with NVTI-06-08 vs homologous boost with a third dose of BBIBP-CorV in 3 groups based on the time interval since administration of 2 doses of BBIBP-CorV at 4-6 months, 7-9 months and >9 months prior to enrollment.

The results are noteworthy and likely to be of interest given the finding of increased neutralizing antibody titers in participants boosted with the recombinant, tri-lineage hybrid vaccine compared to boosting with homologous BBIBP-CorV. Boosting with both vaccines was safe and well-tolerated with low rates of AEs. The findings will be significant in the approach of incorporating 3 RBDs derived from different SARS-CoV-2 strains, specifically wild type, Beta and Kappa, and results showing increased neutralizing antibodies following the heterologous booster compared with homologous booster across all variants of concern tested, including Omicron.

The design and statistical analysis appear overall appropriate. The conclusions are sound and the authors acknowledge the limitation of a substantial gender bias in enrollment. The paper is limited by lack of published data from the Phase 3 trial of the efficacy results of a primary series with the BBIBP-CorV COVID-19 vaccine; the authors reference Phase 1/2 results only and a search did not locate Phase 3 published results. The authors should summarize efficacy, safety and immunogenicity from the Phase 3 trial with BBIBP-CorV to frame waning immunity from the primary series and frame the implication of these results for BBIBP-CorV vaccination and this approach to boosting.

The manuscript is lacking from information of incident covid-19 between enrollment and day 30 post-booster, which is relevant in interpreting immunogenicity data and excluding impact of incident infection if varied across the arms. The randomization design should limit this, but the authors should present the data or comment as a limit if not available.

Abstract:

- Line 45 – would be more specific than “was no less than” or “considerable level” in stating the immunogenicity findings here, and state whether was a statistically significant finding in the abstract.

Introduction

- Line 63 – would update, as breakthrough infections are not growing rapidly in all settings.

- Line 73-74. Provide reference.

- Line 78 – This is repetitive and would delete, and possibly start with, “An effective broad reactive vaccine....”

- Paragraph (line 78+) – would summarize efficacy and immunogenicity data available from the Phase 3 study of the initial series of 2 doses of BBIBP-CorV to frame your findings and cite Phase 3 publication if published.

Methods

- Provide dates of study enrollment
- Please include a statement on informed consent from participants
- Describe how and when participants were tested for incident covid-19 infection on the study and if there were any differences in rates of covid-19 between those receiving the heterologous vs homologous booster during the 30 days post booster when immunogenicity was evaluated.
- Or state whether incident PCR- confirmed infection after vaccination excluded individuals from immunogenicity testing
- Include where Covid variants were procured from in methods for immunogenicity assessments.

Results

- Provide gender breakdown/%male participants in the text given the substantial majority of men enrolled.
- Provide information on predominant variants in participants with PCR-confirmed COVID-19 through day 30 post-booster vaccination if available
- Would replace “top up” with alternate language
- Lines 140-142 – would rework this sentence as cumbersome to read.
- Table 4. Consider narrowing data presented in this table as not clear how much is added by including the ratio of GMC between groups, rate difference between the 2 groups and Ab GMC fold rise. Would move CI for fold rise to brackets on same row to facilitate interpretation.
- Line 168-169. Delete sentence or move to conclusion since is interpretation of data.
- Line 204-211 – rework as difficult to follow.
- Lines 211-213 – delete and address in discussion
- Lines 222-224 – delete and address in discussion

Discussion

- As above, would avoid repetitive statements/conclusions currently in Results section
- Lines 248-250 are repetitive with Lines 253-255 – rework to avoid duplication.
- Lines 267-269 – rework with regard to grammar and flow

- Lines 269-270 – delete.
- Lines 273-274 – delete sentence as repetitive
- Line 275 – delete “obviously”
- Line 277 – replace “top up” with more specific language
- Lines 296-297 – delete sentence as repetitive

Responses to Comments of the First Reviewer:

Reviewer #1 (Remarks to the Author):

Page 10 line 206, these seem like comparison of prototype neutralization versus vs omicron neutralization within each arm (homologous/heterologous). Seems more natural to emphasize the omicron versus prototype difference between arms.

Response: Thank you for your valuable comments. According to the suggestion of the reviewer, in the revised manuscript, the description of the result has been changed to “Both in homologous and heterologous booster groups, the neutralizing antibody level against SARS-CoV-2 Omicron variant was significantly reduced in comparison with that against the prototype strain, indicating the distinct immune-evasive ability of Omicron variant. Our results are consistent with other studies¹⁵⁻¹⁸. However, the anti-Omicron neutralizing titers induced by heterologous boost were still significantly higher than those induced by homologous boost. In participants receiving the homologous boost of BBIBP-CorV, the neutralizing antibody GMT against Omicron variant was 45.03 (95%CI, 36.37-55.74). By comparison, in participants boosted by the heterologous NVSI-06-08, the anti-Omicron neutralizing GMT still maintained at a high level of 367.67 (95%CI, 295.50-457.47), which was 8.17-fold higher than that induced by homologous boost (Fig. 5 and Supplementary Table 5).”. (Please see the details on page 10 highlighted in yellow).

Page 13 line 282. In the discussion you talk about how the benefit of boosting is greater >6 month group compared to the 4-6 month group, presumably from Figure 3a. But I think this is a comparison of increases relative to baseline, and the >9 month group has waned more. The actual D30 neuts look similar in the three groups. Some discussion of this point seems warranted.

Response: Thanks for your insightful and valuable comments. The data presented in the Supplementary Table 3 showed that both the absolute value and the fold rise of post-booster neutralizing antibody titers in the 7-9-month and >9-month groups were significantly greater than those in the 4-6-month group. Especially, although the pre-booster baseline in the >9-month group has waned more, the post-booster neutralizing titers were still distinctly higher than those in the 4-6-month group, as shown in the Supplementary 3. For example, on day 30 after the boost of NVSI-06-08, the neutralizing antibody GMT reached 7719.35 (6828.68-8726.20) and 7479.16 (6540.06-8553.10) in the 7-9-month and >9-month groups, respectively, whereas that in the 4-6-month group was only 3730.18 (3216.15-4326.36). Therefore, both the absolute value of neutralizing GMT and its fold rise relative to baseline demonstrated that the prime-boost vaccination with an interval over 6 months was immunogenically superior to the interval of 4-6 months. The above discussions have been added in the revised manuscript. (Please see the details on page 13 marked in yellow).

Responses to Comments of the Second Reviewer:

Reviewer #2 (Remarks to the Author):

The authors offer a well described study comparing NVS1-06-08 boosting with homologous BBIBP-CoV boosting and give a clear outline of the recruitment, demographics and outcomes. It is a nicely written manuscript and provides valuable insight into new vaccines that induce cross-reactive immunity.

Response: We appreciate the reviewer's positive comments on our manuscript.

I only have minor transcriptional comments

Line 140 Change "Significantly" to "significant increases"

Line 142 change "boosting" to "booster" - change boosting to booster throughout

Line 145 Change "were increased" to "increased by"

Line 148 Change "improved" to "significantly".

Response: Thank you for your careful review of our manuscript. All the grammatical mistakes have been corrected in the revised version.

Fig 3 Neutralising data shows increasing fold rise following the homologous schedule, particularly with the longer intervals. However, with the total IgG (fig 4) there is no change in fold rise with the longer intervals. I wonder if the authors plan to look at the immunoglobulin Fc to see whether this changes over time to account for increasing function (neutralisation) even though total IgG doesn't increase as dramatically. It would be interesting further investigation.

Response: We are deeply grateful for your insightful suggestions. As pointed out by the reviewer, our study showed that longer prime-boost intervals led to higher neutralizing antibody titers. But the IgG antibody levels did not increase with the longer intervals. The results indicated that a wider dose spacing may contribute to the maturation of neutralizing antibodies but may have little effect on non-neutralizing antibodies. The dynamic change tendency was not synchronized between neutralizing and IgG antibodies. As suggested by the reviewer, we will investigate this point in our further study.

Responses to Comments of the Third Reviewer:

Reviewer #3 (Remarks to the Author):

The authors submit a manuscript on a randomized, double blind, controlled trial evaluating a recombinant vaccine incorporating three heterologous RBDs (NVTI-06-080) as a booster vaccine in comparison with homologous booster in individuals who received an initial vaccination series with BBIBP-CorV, an inactivated COVID-19 vaccine. Participants in this study were randomized to either heterologous boost with NVTI-06-08 vs homologous boost with a third dose of BBIBP-CorV in 3 groups based on the time interval since administration of 2 doses of BBIBP-CorV at 4-6 months, 7-9 months and >9 months prior to enrollment.

The results are noteworthy and likely to be of interest given the finding of increased neutralizing antibody titers in participants boosted with the recombinant, tri-lineage hybrid vaccine compared to boosting with homologous BBIBP-CorV. Boosting with both vaccines was safe and well-tolerated with low rates of AEs. The findings will be significant in the approach of incorporating 3 RBDs derived from different SARS-CoV-2 strains, specifically wild type, Beta and Kappa, and results showing increased neutralizing antibodies following the heterologous booster compared with homologous booster across all variants of concern tested, including Omicron.

The design and statistical analysis appear overall appropriate. The conclusions are sound and the authors acknowledge the limitation of a substantial gender bias in enrollment. The paper is limited by lack of published data from the Phase 3 trial of the efficacy results of a primary series with the BBIBP-CorV COVID-19 vaccine; the authors reference Phase 1/2 results only and a search did not locate Phase 3 published results. The authors should summarize efficacy, safety and immunogenicity from the Phase 3 trial with BBIBP-CorV to frame waning immunity from the primary series and frame the implication of these results for BBIBP-CorV vaccination and this approach to boosting.

Response: We appreciate the reviewer's positive comments on our manuscript. As suggested by the reviewer, the efficacy, safety and immunogenicity of BBIBP-CorV from the phase 3 trial were summarized in the revised manuscript. The waning of neutralizing antibodies over time in BBIBP-CorV recipients was also discussed. (Please see the details on page 4 highlighted in yellow).

The manuscript is lacking from information of incident covid-19 between enrollment and day 30 post-booster, which is relevant in interpreting immunogenicity data and excluding impact of incident infection if varied across the arms. The randomization design should limit this, but the authors should present the data or comment as a limit if not available.

Response: Thank you for your insightful and valuable comments. In the trial, nasopharyngeal swabs were collected from all the participants for PCR tests prior to booster vaccination. After the vaccination, only for the subjects who showed any symptoms of COVID-19 or went to hospital for treatment, nasopharyngeal swabs were

collected and PCR tests were conducted. The description on PCR test was added into the “Methods” section of the revised manuscript. (Please see the details on page 17 highlighted in yellow). Results of the study showed that within 30 days after booster vaccination, a total of 23 participants with suspected symptoms were tested by PCR, but none was positive. This result was added into the “Results” section of the revised manuscript. (Please see the details on page 7 marked in yellow). In the trial, the PCR test was not conducted at regular intervals for all the participants, which was discussed as a limitation in the revised manuscript. (Please see the details on page 14 marked in yellow).

Abstract:

- Line 45 – would be more specific than “was no less than” or “considerable level” in stating the immunogenicity findings here, and state whether was a statistically significant finding in the abstract.

Response: Thank you for your valuable suggestions. The non-specific expressions pointed out by the reviewer have been modified in the revised manuscript.

Introduction

- Line 63 – would update, as breakthrough infections are not growing rapidly in all settings.
- Line 73-74. Provide reference.
- Line 78 – This is repetitive and would delete, and possibly start with, “An effective broad reactive vaccine...”
- Paragraph (line 78+) – would summarize efficacy and immunogenicity data available from the Phase 3 study of the initial series of 2 doses of BBIBP-CorV to frame your findings and cite Phase 3 publication if published.

Response: We deeply appreciate the reviewer’s valuable suggestions.

As suggested by the reviewer, the statement “breakthrough infections are growing rapidly.” has been changed to “the reports of breakthrough infections are growing.” (Please see page 3 marked in yellow).

The related reference was provided in the revised manuscript as suggested by the reviewer.

According to the reviewer’s suggestion, the repetitive description was deleted in the revised paper.

According to the reviewer’s suggestion, the efficacy, safety and immunogenicity of BBIBP-CorV from the phase 3 trial were summarized, and the related paper was cited in the revised manuscript. The waning of neutralizing antibodies over time in BBIBP-CorV recipients was also discussed. (Please see the details on page 4 highlighted in yellow).

Methods

- Provide dates of study enrollment

- Please include a statement on informed consent from participants
- Describe how and when participants were tested for incident covid-19 infection on the study and if there were any differences in rates of covid-19 between those receiving the heterologous vs homologous booster during the 30 days post booster when immunogenicity was evaluated.
- Or state whether incident PCR- confirmed infection after vaccination excluded individuals from immunogenicity testing
- Include where Covid variants were procured from in methods for immunogenicity assessments.

Response: We are grateful for your valuable suggestions.

The dates of study enrollment have been added in the revised manuscript. (Please see the details on pages 4 and 14 highlighted in yellow).

The statement “Written informed consent was obtained from all participants before the screening.” has been added to the revised manuscript. (Please see page 15 marked in yellow).

In the trial, nasopharyngeal swabs were collected from all the participants for PCR tests prior to booster vaccination. After the vaccination, only for the subjects who showed any symptoms of COVID-19 or went to hospital for treatment, nasopharyngeal swabs were collected and PCR tests were conducted. The description on PCR test was added into the “Methods” section of the revised manuscript. (Please see the details on page 17 highlighted in yellow). Results of the study showed that within 30 days after booster vaccination, a total of 23 participants with suspected symptoms were tested by PCR, but none was positive. This result was added into the “Results” section of the revised manuscript. (Please see the details on page 7 marked in yellow). In the trial, the PCR test was not conducted at regular intervals for all the participants, which was discussed as a limitation in the revised manuscript. (Please see the details on page 14 marked in yellow).

SARS-CoV-2 viruses used in the live-virus neutralization assays, including the prototype (QD-01), Alpha (BJ-210122-14), Beta (GD84), Delta (GD96) and Omicron (NPRC2.192100003) strains, were provided by the National Institute for Viral Disease Control and Prevention, Chinese Center for Disease Control and Prevention (China CDC). This information was included in the revised manuscript. (Please see page 19 highlighted in yellow).

Results

- Provide gender breakdown/%male participants in the text given the substantial majority of men enrolled.
- Provide information on predominant variants in participants with PCR-confirmed COVID-19 through day 30 post-booster vaccination if available
- Would replace “top up” with alternate language
- Lines 140-142 – would rework this sentence as cumbersome to read.
- Table 4. Consider narrowing data presented in this table as not clear how much is added by including the ratio of GMC between groups, rate difference between the 2

groups and Ab GMC fold rise. Would move CI for fold rise to brackets on same row to facilitate interpretation.

- Line 168-169. Delete sentence or move to conclusion since is interpretation of data.
- Line 204-211 – rework as difficult to follow.
- Lines 211-213 – delete and address in discussion
- Lines 222-224 – delete and address in discussion

Response: Thank you for your careful review of our manuscript.

As suggested by the reviewer, the proportion of male participants was provided in the main text of the revised manuscript. (See page 5 marked in yellow).

According to the study protocol, a total of 23 participants with suspected symptoms were tested by PCR within 30 days post-booster, but all the test results were negative. Therefore, no information about the predominant variants of infections was obtained.

The word “top up” has been changed to “improve” in the revised paper.

The sentence “Significantly increase in neutralizing antibody titers against the prototype SARS-CoV-2 virus, detected by live-virus neutralization assays, were observed after both homologous and heterologous boosting vaccinations compared to the pre-boosting baseline values.” has been modified as “Immunogenicity analysis showed that both homologous and heterologous booster vaccinations significantly improved the neutralizing antibody titers against the prototype SARS-CoV-2 virus.” (Please see page 7 highlighted in yellow).

In the new version of the manuscript, the Supplementary Tables 3 and 4 were revised according to the reviewer’s suggestions. The 95% CI for the ratio of 4-fold rise was moved to brackets on the same row.

According to the suggestion of the reviewer, the sentence that interprets the trial data was removed from the Results section.

The unclear descriptions in the manuscript as pointed out by the reviewer have been revised for more clarity. (Please see the details on page 10 highlighted in yellow).

As suggested by the reviewer, the sentences that interpret the trial data have been removed from the Results to Discussion sections.

Discussion

- As above, would avoid repetitive statements/conclusions currently in Results section
- Lines 248-250 are repetitive with Lines 253-255 – rework to avoid duplication.
- Lines 267-269 – rework with regard to grammar and flow
- Lines 269-270 – delete.
- Lines 273-274 – delete sentence as repetitive
- Line 275 – delete “obviously”
- Line 277 – replace “top up” with more specific language
- Lines 296-297 – delete sentence as repetitive

Response: We are grateful for the careful review of our manuscript.

As suggested by the reviewer, the repetitive sentences were deleted, and the

grammatical mistakes were corrected in the revised version of the manuscript. The details are as follows:

(1) The sentence “The antigen of NVSI-06-08 was designed to realize trimerization of RBDs without introducing any exogenous sequence, which facilitated the cross-links of B cell receptors but without introducing additional safety risks.” was changed to “The antigen of NVSI-06-08 was designed without introducing any exogenous sequence, which did not bring additional safety risks.” (Please see page 12 marked in yellow).

(2) The sentence “NVSI-06-08 was designed as a hybrid-type COVID-19 vaccine with broad protection potential, which integrated multiple antigens into a single molecule without introduction any exogenous linker.” was changed to “NVSI-06-08 was designed as a hybrid-type COVID-19 vaccine with broad protection potential, which integrated multiple antigens into a single molecule.” (Please see page 12 highlighted in yellow).

(3) The sentence “We think that update of the vaccine to incorporate the Omicron-carrying mutations into the immunogen should induce better immunogenicity against Omicron variant.” was changed to “We believe that updating the vaccine by including Omicron-carrying mutations into the immunogen should induce better immunogenicity against Omicron variant.” (Please see page 13 highlighted in yellow).

(4) The sentence “This strategy has been recently validated in animal experiments by our group (data not shown).” was deleted.

(5) The sentence “Our studies provide a promising method for broad-spectrum COVID-19 vaccine development.” was deleted.

(6) The word “obviously” was deleted.

(7) The word “top up” was changed to “enhance”.

(8) The sentence “Our study provides a preferred strategy to top up the immunity in BBIBP recipients against SARS-CoV-2 and its variants.” was deleted.